# Experimental Study on Static Mechanical Properties and Moisture Contents of Concrete Under Water Environment



**Guohui Zhang [1] , Xiaohang Li [1] and Zongli Li [2,***

[1]   Electrical Engineering College, Kunming University of Science and Technology,
     Kunming 650500, Yunnan, China; zgh_water@kumust.edu.cn (G.Z.); lxh@kumust.edu.cn (X.L.)
[2]   College Water Resources & Architectural Engineering, Northwest A&F University,
     Yangling 712100, Shaanxi, China
*   Correspondence: bene@nwsuaf.edu.cn; Tel.: +86-1366-976-7383

**Abstract:** This paper presents an experiment to investigate the influence of moisture on the static mechanical properties of concrete, and prediction equations for strength and fracture toughness of concrete at different strength grades, relative to water saturation, were established respectively. The research results show that all of the compressive strength, splitting tensile strength, and fracture toughness of concrete exhibit an approximately linearly decreasing trend with the increase in water saturation. For saturated concrete specimens with w/c 0.65, 0.55, 0.42 compared with dry ones, compressive strength decreases by 40.08%, 36.08%, and 33.73%, respectively, splitting tensile strength decreases by 45.39%, 42.61%, and 35.18%, respectively, and fracture toughness decreases by 57.31%, 49.92%, and 46.76%, respectively. The higher the water saturation of concrete, the larger the slope of the ascending part of the uniaxial compressive stress-strain curve, and the smaller the peak strain corresponding to the peak compressive stress, then in this case, both crack mouth opening displacement and loading point deflection corresponding to the critical load on three-point bending beam, decrease. Ingress of water causes the deformation capacity to decrease, and the toughness to weaken, which are unfavorable to the mechanical properties of concrete.

**Keywords:** concrete; moisture content; compressive strength; splitting tensile strength; fracture toughness

## 1. Introduction

For such concrete structures as dams, foundation and pier of a river-crossing bridge, coastal structures, and offshore production platforms, part or all of their surfaces are often in contact with water. Concrete has a large quantity of capillaries and pores, therefore water may permeate into the pores and cracks of the concrete, so that the concrete has different moisture contents, and as shown by studies [1–4], the existence of water causes significant change in the mechanical properties of concrete. Strength and fracture toughness directly relates to the reliability of concrete structure projects, so it is especially important to systematically study the regularities of the mechanical properties of concrete, as it varies with different moisture contents. The research efforts on the influence of water on the mechanical properties of concrete mainly relate to the chemical effect and mechanical effect of water at present, and this paper studies the change in static mechanical properties of concrete induced by the mechanical effect of water [5,6].

Reinhard et al. [7] experimentally studied the regularities of the influence of water on the mechanical properties of concrete, and obtained that when the strain rate was 0.50 s$^{-1}$, the tensile strength of saturated concrete was only 75% of that of dry concrete. The experimental results obtained

by Ross et al. [8] also reveal that both the compressive strength and tensile strength of concrete decreased somewhat with any increasing moisture content at a low loading rate. Wittmann et al. [9] baked concrete specimens at 105 °C for 14 days to get dry concrete specimens, and then placed the dry concrete specimens into water for different immersion times, to obtain concrete specimens with water saturations of 75%, and others with water saturation of 100%; thereafter they tested the compressive strength of the specimens. The research results show that, compared with dry concrete, the concrete specimens with the water saturation of 75% and those with a water saturation 100%, had compressive strengths decreasing by 17.45% and 33.79%, respectively. Li [10] found through experimental study that the strength of dry concrete was evidently higher than that of water-immersed concrete, decreasing gradually with the increase in immersion time, and the influence of moisture content on the strength of concrete with a high water-cement ratio was more evident than on the strength of concrete with a low water-cement ratio. In a study, Cadoni et al. [11] baked concrete specimens at 50 °C until dry, and then placed the dry concrete specimens into water for immersion; afterwards, the concrete specimens were taken out for a strength test. The results show that the tensile strength of concrete decreased from 3.28 MPa in the dry state to 3.02 MPa in its saturated state, with a decrease magnitude of 7.62%. Wang and Li [12,13] placed concrete specimens into water and cured them for 48 days to get saturated concrete specimens, and baked the specimens at 50 °C for 2 days, and at 65 °C for 6 days, so that the specimens reached a dry state. The research results indicate that saturated concrete has a compressive strength decreasing by 4.5%, and splitting tensile strength decreasing by 11.41%, compared with dry concrete. After Yan et al. [14] immersed concrete specimens with an age of 300 days into water for 60 days in their study, the specimens had moisture content increasing from 0.31% to 4.80%, and tensile strength decreasing by 41.2%. Deng et al. [15] studied the regularities for sandstone fracture toughness changing under a water immersion effect, and found that the fracture toughness decreased with the increase in immersion time, and the decrease magnitude had a trend of increasing first, and then decreasing. The experiment by Rossi [16,17] revealed that the fracture toughness of saturated concrete at the low loading rate decreased somewhat, compared with that of dry concrete, but the fracture toughness values obtained under a saturated condition in this experiment have relatively large discreteness. Wittmann et al. [18] and Zhang et al. [19] studied concrete, and obtained that fully dry concrete had the highest fracture energy, which was 1.18 times the fracture energy of concrete with a water saturation of 75%, and 1.57 times the fracture energy of concrete with a water saturation of 100%, and the deformation capacity of concrete weakened gradually with increasing moisture content.

Most of the current research efforts on the influence of moisture content on the mechanical properties of concrete only relate to a single mechanical property, e.g., compressive strength, or tensile strength, and set the moisture content mainly as a dry or saturated state, lacking of study on the regularities of the mechanical properties of concrete changing at other different moisture contents. Different researchers use different drying methods when drying concrete specimens to determine the moisture content, and the difference in strength due to different drying methods will inevitably interfere with the evaluation of the degree of influence of moisture content on the strength of concrete. This research team carried out a study on the influence of any drying condition on the strength of concrete [20], and got the drying process with the minimum damage to concrete. Based on this drying method, the free water absorption process of concrete was studied first in this paper to generate concrete specimens with different moisture contents, and then the regularities of compressive strength, splitting tensile strength, and the fracture toughness of concrete specimens changing at three strength grades were systematically studied at six different moisture contents.

## 2. Materials and Methods

### 2.1. Test Materials

In this investigation, a complex Portland Cement (P.C32.5) produced in the Jidong cement plant of China was used in all formulations, and all its properties were in accordance with Chinese standard

GB175-2007 [21]. The water was obtained directly from the city drinking water supply. The aggregate included medium sand (fine aggregate) and pebbles (coarse aggregate) produced in the Weihe plant of Yangling in Shaanxi Province, and the fineness modulus, clay content, apparent density and bulk density of the sand were 2.43, 0.8%, 2590 kg/m$^3$, and 1540 kg/m$^3$ respectively, while the clay content, apparent density, bulk density and maximum aggregate size of the pebbles were 0.6%, 2650 kg/m$^3$, 1563 kg/m$^3$, and 40 mm, respectively. The strength levels of concrete in this research were C15, C20 and C30, which were cured under the standard conditions (20 °C ± 2 °C, RH > 95%) for 28 days. The mix proportions of concrete used are shown in Table 1 according to Chinese standard DL/T5330-2005 [22]. It is worth noting that the strength grade of concrete involved in this research is the strength grade in Chinese code.

**Table 1.** Mix proportion of concrete.

| Strength Grade | | Mix Proportion/(kg·m$^{-3}$) | | | | |
|:---:|:---:|:---:|:---:|:---:|:---:|:---:|
| **China** | **European** | **w/c** | **Water** | **Cement** | **Fine Aggregate** | **Coarse Aggregate** |
| C15 | C12 | 0.65 | 158 | 243 | 729 | 1418 |
| C20 | C16 | 0.55 | 150 | 273 | 615 | 1370 |
| C30 | C25 | 0.42 | 165 | 391 | 581 | 1292 |

*2.2. Test Procedure*

2.2.1. Water Saturation Test Procedure

Standard cube specimens were used, and their dimensions were 150 × 150 × 150 mm. The experimental control factors were strength grade and water immersion time; three experimental groups A, B, and C with strength grades of C15, C20, and C30, respectively, were set up, each group contained three standard specimens, and there were nine specimens in total. After being cured under standard conditions for 28 days, the specimens were placed into an electric thermostatic drying oven with forced air convection respectively, and baked at 105 °C, with air blown continuously until the specimens had constant mass, reaching a fully dry state. After the specimens cooled down naturally, the mass of each fully dry specimen was recorded; and then, the specimens were placed into water tanks in groups, and tap water was added slowly, so that the water surface was flush with the top surfaces of the specimens, and the specimens were ensured to be still immersed after absorbing water. Within 10 h after the start of immersion, the specimens in each group were taken out and wiped with a wet cloth once per 0.5 h, and then the mass of each immersed specimen was measured; afterwards, the specimens were placed back into the tanks for continuous immersion. Within 10–20 h of immersion time, the mass of each specimen was measured once per hour; with the elapse of immersion time, the time interval for measuring specimen mass was increased properly; the total immersion time was 270 h for all of these three experimental groups A, B, and C. Through intermittently recording the change in mass of each specimen with immersion time in these three experimental groups A, B, and C, and calculating the water saturation of the specimen with Equation (1), it was determined that when the water absorption rate (water amount absorbed by specimen within unit time) of concrete specimen is less than 0.1 g/h, the specimen is regarded to reach approximately saturated state.

$$s_i = \frac{m_i - m_o}{m_w - m_o} \times 100\% \tag{1}$$

where $s_i$ denotes the modulus water saturation of concrete (%); $m_o$ represents the mass of the concrete specimens dried completely (kg); $m_w$ represents the mass of the concrete specimens saturated (kg); $m_i$ represents the mass of the concrete specimens measured under different moisture conditions (kg).

### 2.2.2. Mechanical Properties Test Procedure

The experimental control factors were the strength grade and water saturation of concrete, and the experimental materials and mix proportions were the same as in the experimental groups A, B, and C in Section 2.2.1 of this paper. Eighteen experimental groups were set up according to the preset factors, and six experimental groups were set up for each strength grade, as shown in detail in Table 2; each group had 11 specimens, and there were 198 specimens in total; the types, quantity and dimensions of the specimens in every experimental group are shown in Table 3.

**Table 2.** Test group serial number.

| Strength Grade | Dry Group | Immersion Time/h | | | | |
|---|---|---|---|---|---|---|
| | | 3 | 10 | 24 | 72 | 240 |
| C15 | $A_0$ | $A_1$ | $A_2$ | $A_3$ | $A_5$ | $A_7$ |
| C20 | $B_0$ | $B_1$ | $B_2$ | $B_3$ | $B_5$ | $B_7$ |
| C30 | $C_0$ | $C_1$ | $C_2$ | $C_3$ | $C_5$ | $C_7$ |

**Table 3.** Specific content of each group.

| Specimen Type | Amount | Size/mm | Text Item |
|---|---|---|---|
| Standard cube | 3 | $150 \times 150 \times 150$ | Compressive strength |
| Standard cube | 3 | $150 \times 150 \times 150$ | Splitting tensile strength |
| Three-point bend beam | 5 | $100 \times 100 \times 515$ | Fracture toughness |
| Summation | 11 | | |

The fracture toughness was determined with the precracked three-point bending beam concrete specimens shown in Figure 1, and each specimen had a length (L) of 515 mm, a thickness (t) of 100 mm, a height (h) of 100 mm, and a span (S) of 400 mm, with single-edge crack prefabricated, which had a crack angle of 0°, and a crack tip opening angle of 15°. The prefabricated crack had a depth of $a_0 = 50$ mm, a width of 3 mm, and the ratio of the crack depth to the specimen height was $a_0/h = 0.5$. The prefabricated crack was formed through an embedded steel plate. Before concrete pouring, both sides of a steel plate were applied with mold release agent; after initial setting of the concrete in 3 h, the steel plate was pulled out to form a prefabricated crack.

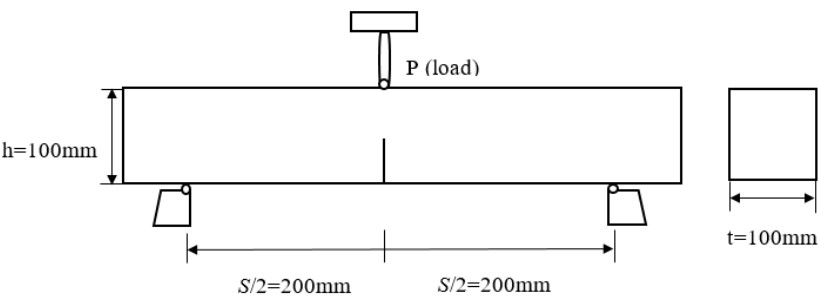

**Figure 1.** Configuration of three-point bending precracked beam.

$A_0$, $B_0$, and $C_0$ were the dry groups of the C15, C20, and C30 concrete specimens, respectively; after the specimens reached dry state, out of 11 specimens in each experimental group, three standard cube specimens were used for the test of the compressive strength of concrete, three other standard cube specimens were used for the test of the splitting tensile strength of concrete, and the remaining five three-point bending beam specimens were used for the fracture test, so as to obtain the compressive strength, splitting tensile strength, and fracture toughness of concrete specimens at three strength grades, i.e., C15, C20, and C30, in dry state. $A_1$ through $A_5$, $B_1$ through $B_5$, and $C_1$ through $C_5$

were experimental groups of C15, C20, and C30 concrete specimens with different water saturations, respectively. The specimens in each group were caused to reach their dry state first, using the same drying method as that for groups $A_0$, $B_0$, and $C_0$. After the specimens cooled down naturally, the mass of each dried specimen was recorded. After this, the specimens were placed into water tanks in groups, and tap water was added slowly, so that the water surface was flush with the top surfaces of the specimens, and the specimens were ensured to be still immersed after absorbing water (the water depth was 150 mm, so the static water pressure could be neglected). Five experimental groups $A_1$ through $A_5$ were immersed for 3 h, 10 h, 24 h, 72 h, and 240 h, respectively; and then the specimens were taken out from the water tanks, and the mass of each immersed specimen was measured. For each group of immersed specimens, tests of the compressive strength of concrete, the splitting tensile strength of concrete, and three-point bending beam fracture were carried out, respectively; and for the other ten experimental groups, namely, $B_1$ through $B_5$ and $C_1$ through $C_5$, tests of strength and fracture toughness were done using the same method.

Both compressive strength and splitting tensile strength tests of concrete were carried out using a microcomputer-controlled fully automatic pressure tester belong to Kunming University of Science and Technology, which was manufactured by Shanghai Xinsasi Measuring Instrument & Equipment Manufacturing Co., Ltd., China, whose model number was YAW4206, and whose maximum test force was 2000 kN. The loading rate was 0.3 MPa/s for the test of compressive strength and 0.05 MPa/s for the test of splitting tensile strength, and the concrete strength test was conducted according to the Hydraulic Concrete Test Procedure (SL352-2006) [23]. The fracture toughness test was carried out by a fracture test of a three-point bending beam, and the three-point bending beam test of concrete was carried out using a microcomputer-controlled electro-hydraulic servo universal tester, whose model number was SHT4305, and whose maximum test pressure was 300 kN. The load and displacement were measured using load cells and displacement transducers, respectively, the crack mouth opening displacement (*CMOD*) was measured using a clip-on extensometer, and a computer acquisition system connected with the tester was used to record and display data, and to automatically record the whole process, curves of load versus loading point deflection, and of load versus *CMOD* in real time. In the three-point bending beam test, an equal displacement rate was used to control loading, and the loading rate was 0.1 mm/min; the test process was conducted strictly according to the Hydraulic Concrete Fracture Test Procedure (DL/T5332-2005) [24].

As stipulated in the Chinese national standard SL352-2006 (Hydraulic Concrete Test Procedure), the compressive strength should be calculated as the arithmetic mean of the measured value of three specimens. Besides, in case the difference between the maximum or minimum compressive strength of one specimen and the intermediate compressive strength of three specimens is 15% over the intermediate compressive strength, the intermediate compressive strength of these three specimens should be taken as the value of the compressive strength; The test will be invalid in the event of two specimens obeying the above provision, and therefore at this time, this experiment must be repeated. The relative compressive strength is defined as the ratio of concrete compressive strength under different drying conditions to that of under the standard condition.

## 3. Results

### 3.1. Water Saturation and Immersion Time

Figure 2 shows the curves of water saturation versus immersion time for C15, C20, and C30 concrete specimens, and it can be seen from Figure 2 that, for all experimental groups A, B, and C, the water saturation increased with the increase in immersion time. When the immersion time was 0–72 h, the lower the strength grade, the faster the increase in speed of the water saturation, and under the same immersion time condition, the lower the strength grade, the higher the water saturation; for example, when the immersion time was 24 h, the water saturation of C15 concrete was 1.09 times that of C20 concrete and 1.17 times that of C30 concrete. When the immersion time reached 72 h, the water

saturations of C15, C20, and C30 had reached 96.88%, 96.59%, and 97.70%, respectively; with a further increase in immersion time, the increase speed of water saturation turned slower. When the immersion time reached 270 h, the average water absorption rate of the specimens was less than 1 g/h in all groups A, B, and C, and the specimens had reached saturated state. The water absorption process of concrete is complex and slow, and absolute saturation cannot be achieved within a short time, so the saturated state defined in this paper is the approximately-saturated state.

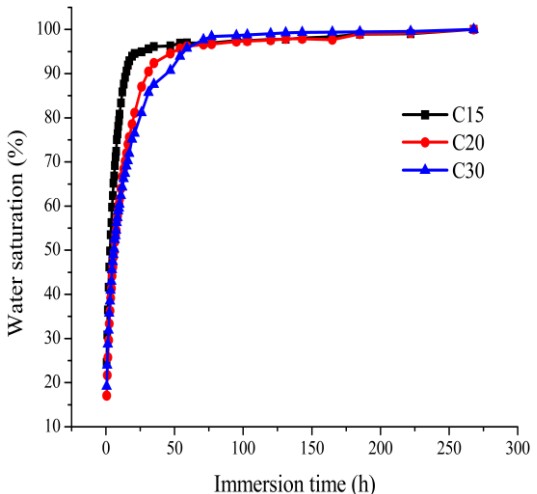

**Figure 2.** The curve of water saturation.

With C15 concrete as an example, the water absorption process of concrete was further analyzed, the change process of the water saturation of concrete was discussed, and the curves of water saturation and water absorption rate of C15 concrete versus immersion time were plotted, as shown in Figure 3. As can be seen from Figure 3, the water absorption process of concrete was roughly divided into three stages, i.e., AB, BC, and CD. Stage AB: This is a linear water absorption stage with a time interval of about 0–3 h, where the water saturation increased rapidly with a linear relationship, the average water absorption rate was 56.7 g/h, and the water saturation of specimens immersed for 3 h was 49.81%. At this stage, the water primarily entered the pores of dry concrete through adsorption and surface diffusion, and formed adsorption layers on the pore walls, and the water entered inside the specimens through diffusion, exhibiting surface water absorption. Stage BC: This is a nonlinear water absorption stage, wherein the water saturation exhibited a nonlinear progressive increase, the water absorption rate decreased gradually, and the time interval of this stage was about 3–72 h. With a further increase in internal moisture content, the water was transferred into the specimens continuously in the form of water film, and local water saturation occurred, forming liquid seepage. Stage CD: This is an approximate saturation stage with a time interval of 72–270 h, where the water absorption rate at this stage was extremely slow, and the water saturation increased by only 3.12%.

A little water was transferred into the specimens continuously in the form of water film, forming liquid seepage; in addition, under the effect of the moisture content gradient, the water diffused gradually from the outer layer to the inner layer, and the water was distributed into the whole concrete specimen gradually; the overall water saturation of specimens at this stage tended to be stable.

### 3.2. Water Saturation and Strength

According to the experimental results of the compressive strength and splitting tensile strength of concrete, statistical analysis was made for the compressive strength and splitting tensile strength values of C15, C20, and C30 concrete specimens from dry state to saturated state, and the strength value of each experimental group was the arithmetic mean of those strength values of the three specimens. To eliminate the experimental errors due to concrete age differences at different immersion times,

the compressive strength and splitting tensile strength values of concrete aged over 28 days were uniformly converted into strength values of concrete aged 28 days according to references [25], and no conversion was made for concrete with age less than 1 day more than 28 days. Prediction equations for compressive strength and splitting tensile strength of concrete relative to water saturation were fitted using the method of least squares, as shown in Equations (2) and (3). Figure 4; Figure 5 are comparative diagrams between the experimental values and fitted curves of the compressive strength and splitting tensile strength of C15, C20, and C30 concrete specimens versus water saturation.

$$f_{ci} = KS_i + B \tag{2}$$

$$f_{ti} = K'S_i + B' \tag{3}$$

where $f_{ci}$ represents the compressive strength of the concrete under different water saturation (MPa); $f_{ti}$ represents the splitting tensile strength of the concrete under different water saturation (MPa); $s_i$ denotes the modulus water saturation of concrete (%); $K$, $K'$, $B$ and $B'$ are fitted parameters given in Table 4.

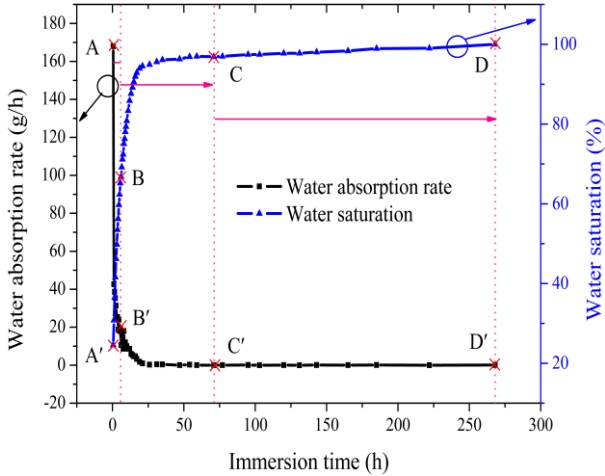

**Figure 3.** The curve of water saturation and water immersion time absorption rate.

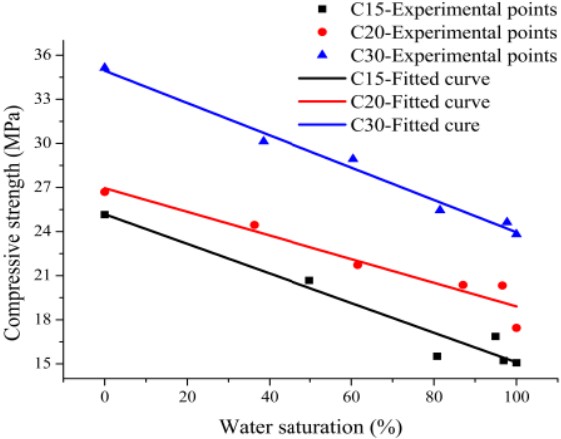

**Figure 4.** Comparative diagrams between the experimental values and fitted curves of splitting tensile strength.

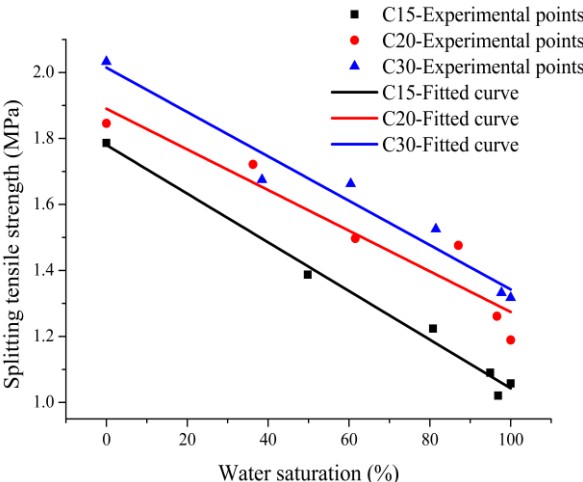

**Figure 5.** Comparative diagrams between the experimental values and fitted curves of the compressive strength.

**Table 4.** Fitting parameters.

| Strength Grade | Compressive Strength | | | Splitting Tensile Strength | | |
|---|---|---|---|---|---|---|
| | **K** | **B** | $R^2$ | $K'$ | $B'$ | $R^2$ |
| C15 | −0.101 | 25.181 | 0.94 | −0.007 | 1.780 | 0.99 |
| C20 | −0.081 | 26.955 | 0.91 | −0.006 | 1.890 | 0.90 |
| C30 | −0.110 | 34.945 | 0.98 | −0.007 | 2.015 | 0.94 |

As can be seen from Figures 4 and 5, the experimental values and fitted curves of the compressive strength and splitting tensile strength of concrete versus water saturation fit well, the fitting Equations (2) and (3) have a good correlation, and can be used to predict the strengths of concrete at different water saturations. Both compressive strength and splitting tensile strength of C15, C20, and C30 concrete specimens exhibit an approximately linearly decreasing trend with the increase in water saturation. Given the same water saturation, the lower the strength grade of concrete, the larger the decrease magnitudes of compressive strength and splitting tensile strength. For example, when all concrete specimens were in their water-saturated state, the compressive strengths of C15, C20, and C30 concrete specimens decreased by 40.08%, 34.65%, and 32.23%, respectively, and the splitting tensile strengths decreased by 40.80%, 35.57%, and 35.17%, respectively, compared with the experimental groups in the dry state. Wittman et al. [9] found through study that the compressive strength of saturated concrete decreased by 33.79%, compared with dry concrete, and Reinhard et al. [7] obtained through study that the tensile strength of saturated concrete was only 75% of that of dry concrete; this paper makes conclusions consistent with the those above, but the decrease magnitudes differ somewhat from the above conclusions due to the different definitions of dry state and approximately water-saturated state of concrete, and different experimental conditions, including strength level, aggregate type, and mix proportion.

With the C20 concrete specimen groups ($B_2$, $B_4$, and $B_5$) as examples, the regularities of influence of the water saturation on the uniaxial compressive stress-strain relationship of concrete were analyzed; only the ascending part and partial descending part of the stress-strain relationship curves were obtained, due to the limitations of experimental instruments; Figure 6 gives the uniaxial compressive stress-strain relationship curves of concrete versus water saturation. As can be seen from Figure 6, the higher the water saturation of concrete, the larger the slope of the ascending part of the stress-strain curve and the smaller the peak stress, indicating that any ingress of water causes the elastic modulus of concrete to increase, and the compressive strength to decrease. The higher the water saturation, the smaller the peak strain ($\varepsilon_c$) corresponding to peak stress; for example, the peak strain of water-saturated concrete specimens in experimental group ($B_5$) was only 80.78% of that of concrete specimens with a

water saturation of 61.55% in experimental group ($B_2$)—in other words, the increase in water saturation caused the concrete to become brittle, and its deformation capacity to decrease.

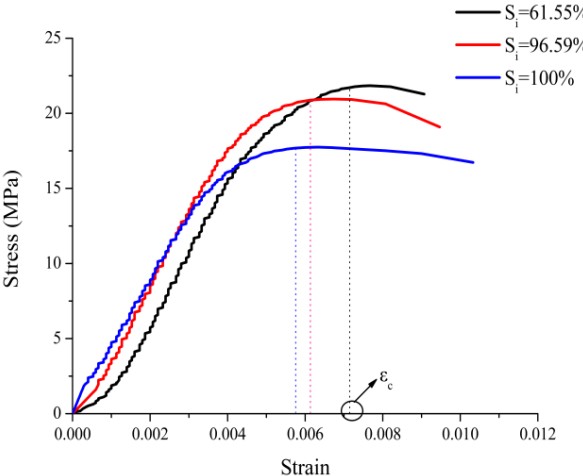

**Figure 6.** Uniaxial compression stress-strain relationship.

*3.3. Water Saturation and Fracture Toughness*

The fracture toughness values at different water saturations were calculated using Equation (4) given by the American Society for Testing and Materials (ASTM), and the critical load and fracture toughness values at different water saturations are shown in Table 5.

$$K_{IC} = \frac{P_{max} - S}{th^{3/2}} f_1\left(\frac{a_0}{h}\right), \tag{4}$$

$$f_1\left(\frac{a_0}{h}\right) = 2.9\left(\frac{a_0}{h}\right)^{1/2} - 4.6\left(\frac{a_0}{h}\right)^{3/2} + 21.8\left(\frac{a_0}{h}\right)^{5/2} - 37.6\left(\frac{a_0}{h}\right)^{7/2} + 38.7\left(\frac{a_0}{h}\right)^{9/2}$$

where $K_{IC}$ denotes the fracture toughness of the concrete at different water saturation (MPa·m$^{1/2}$); $P_{max}$ represents the maximum load of the concrete (N).

**Table 5.** Fracture toughness of different water saturation.

| | C15 | | | C20 | | | C30 | |
|---|---|---|---|---|---|---|---|---|
| NO. | $P_{max}/$ N | $K_{IC}/$ MPa·m$^{1/2}$ | NO. | $P_{max}/$ N | $K_{IC}/$ MPa·m$^{1/2}$ | NO. | $P_{max}/$ N | $K_{IC}/$ MPa·m$^{1/2}$ |
| $A_0$ | 1500 | 0.506 | $B_0$ | 1909 | 0.643 | $C_0$ | 2203 | 0.742 |
| $A_1$ | 1047 | 0.353 | $B_1$ | 1435 | 0.484 | $C_1$ | 1867 | 0.629 |
| $A_2$ | 939 | 0.316 | $B_2$ | 1347 | 0.454 | $C_2$ | 1667 | 0.562 |
| $A_3$ | 760 | 0.256 | $B_3$ | 1147 | 0.387 | $C_3$ | 1416 | 0.477 |
| $A_4$ | 717 | 0.242 | $B_5$ | 1058 | 0.357 | $C_5$ | 1269 | 0.428 |
| $A_5$ | 642 | 0.216 | $B_7$ | 956 | 0.322 | $C_7$ | 1172 | 0.395 |

According to the experimental data shown in Table 5, prediction Equations (5), (6) and (7) for fracture toughness of C15, C20, C30 concrete relative to water saturation were fitted, and the correlation coefficients were 0.93, 0.99, and 0.97, respectively.

$$K_{IC} = -0.003S_i + 0.500 \tag{5}$$

$$K_{IC} = -0.003S_i + 0.643 \tag{6}$$

$$K_{IC} = -0.004S_i + 0.752 \tag{7}$$

Figure 7 shows the comparative diagram between the experimental data of fracture toughness of concrete at different water saturations and the fitted curves. It can be seen from Figure 7 that the prediction equations fit the experimental results well. For C15, C20, and C30 concrete specimens, fracture toughness exhibited an approximately linearly decreasing trend with the increase in water saturation, and the influence effect was significant. For example, the fracture toughness values of water-saturated concrete specimens in experimental groups ($A_5$, $B_5$, and $C_5$) were only 42.69%, 48.92%, and 53.24% of those of fully dry concrete specimens in groups ($A_0$, $B_0$, and $C_0$), respectively. Wittmann et al. [18] obtained through study that the fracture energy of concrete decreased as the water saturation degree of concrete increased, and the fracture energy of water-saturated concrete was only 63.5% of that of dry concrete. Zhang et al. [19] found through study that the fracture energy of water-saturated concrete decreased by 23.9%, compared with that of dry concrete. Because of the limited space of this paper, the fracture, failure and deformation regularities of concrete at different water saturations were analyzed with the groups in the fully dry state, partially-saturated state and fully-saturated state ($C_0$, $C_1$, and $C_5$), out of the experimental groups of C30 concrete as examples, and Figure 8; Figure 9 show the P-CMOD and P-deflection curves of C30 concrete at different water saturations, respectively.

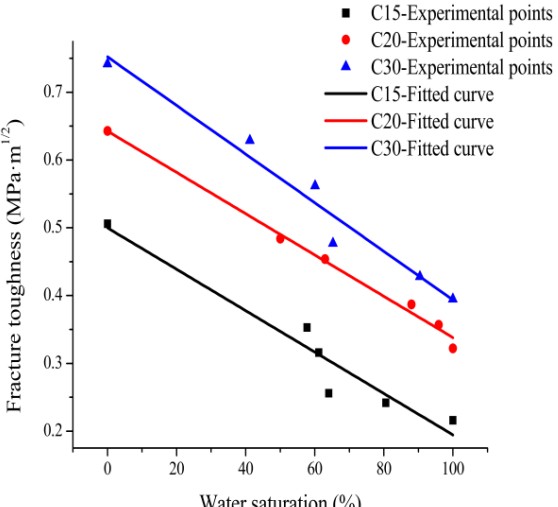

**Figure 7.** Comparative diagrams between the experimental values and fitted curves of fracture toughness.

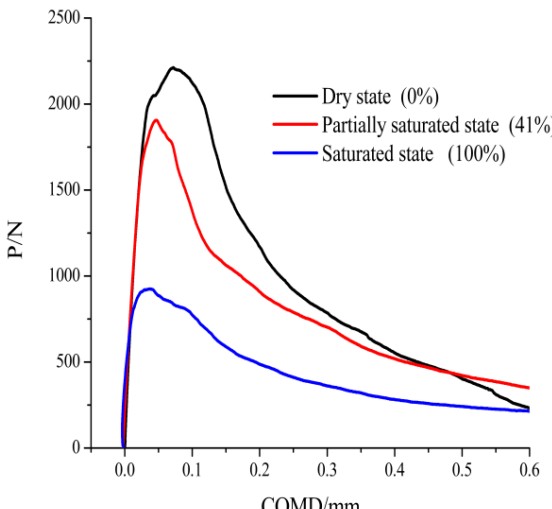

**Figure 8.** The P-CMOD curves of concrete.

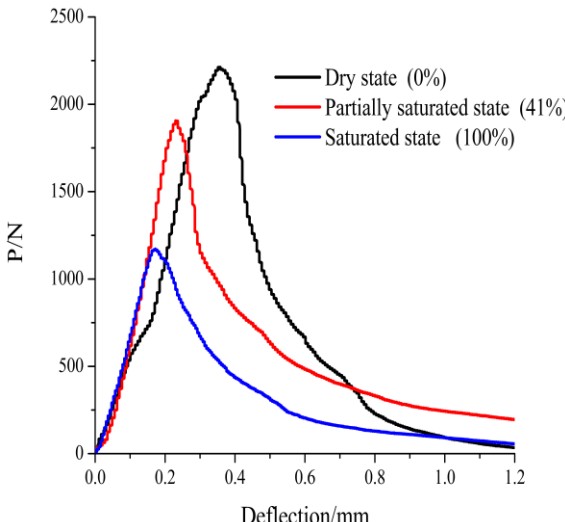

**Figure 9.** The P-deflection curves of concrete.

As can be seen from Figure 8; Figure 9, the slopes of the ascending parts of P-CMOD and the P-deflection curves in the experimental group ($C_0$) of dry concrete are the smallest, and increase gradually with an increasing water saturation. The CMOD and loading point deflection, corresponding to the critical load for fast crack propagation, are written as $COMD_C$ and $Y_C$ in short, respectively, and both $CMOD_C$ and $Y_C$ exhibit a decreasing trend with the increase in water saturation. For example, the $CMOD_C$, and $Y_C$ are 0.030 mm and 0.189 mm, respectively, in the experimental group ($C_5$) in the water-saturated state, and 0.0721 mm and 0.353 mm, respectively, in the experimental group ($C_0$) in dry state, and the $CMOD_C$ and $Y_C$ in the experimental group in water-saturated state decreased by 58.39% and 30.65%, respectively, compared with those in the experimental group in dry state. Under the same CMOD and deflection conditions, the higher the water saturation of a specimen, the smaller the tolerable load; for example, when the CMOD was 0.050 mm, the tolerable load of a specimen was 2072 N in the experimental group ($C_0$) in dry state, whereas the tolerable load was 886 N in the experimental group ($C_5$) in approximately-saturated state, decreasing by 57.2% compared with the load in dry state.

## 4. Discussion

There are many kinds of interpretations for the physical mechanism of decreased strength of wet concrete. Popovics [26], and Bartlett and Macgregor [27] believed that when there is a moisture content difference in concrete, the part with the higher moisture content expands, and the inhomogeneous deformation generates inhomogeneous tensional and compressive stresses in the concrete, so that the concrete strength decreases; however, for the concrete in a fully-saturated state without any moisture content difference in experiment, its strength still decreases greatly, compared with the strength in the fully dry state. Oshita and Tanabe [28,29] found through measuring the change in pore water pressure in concrete that, when the volumetric deformation of wet concrete is in its compression state, the pore water pressure increases gradually, and exhibits a linear relationship with deformation, and it is believed that pore water pressure has a promotion effect on the cracking of concrete. Guo and Waldron [30] calculated the axial compressive stress of concrete using the elastic mechanical method, and found that the maximum circumferential stress of concrete with pores full of water was 67% more than that of concrete with pores full of air, and the circumferential stress may cause the concrete to be destroyed, thus resulting in a decrease in the compressive strength of concrete. However, research results revealed that, in the case that it is hard to generate pore water pressure in wet concrete under tensional load effect, the tensile strength still decreased significantly; so both of the above theories cannot fully reflect the mechanism of the effect of water on the mechanical properties of concrete.

The decrease in strength of wet concrete can be interpreted from the perspective of energy, the work done by external force during the generation, convergence and extension of cracks in concrete needs to overcome the surface energy forming microcracks. After water permeates into concrete, it reduces the van der Waals forces between the microscopic particles of materials, and weakens the cohesion among the particles of concrete, so that the surface energy is reduced, and then the energy needed to form new fracture planes is decreased, and the work done by any external force for extension of initial cracks is less, macroscopically exhibiting a decrease in strength and fracture toughness [18–31]. The other kind of interpretation is that the fracture and failure of concrete start from the interfacial transition zone between aggregate and mortar first, the main hydration product of the cement mortar in the transition zone is C–S–H, whose volume percentage is about 60–70% and whose strength and fracture property directly dictate the macroscopic mechanical properties of concrete. The molecular structure diagrams of C–S–H at different water saturations were given [32], and the mechanism of the damage effect of water absorption rate on the mechanical properties of concrete was interpreted from the perspective of molecular dynamics, as shown in Figure10.

The molecular structure of C–S–H is one layer of calcium oxygen laminar structure sandwiched between two layers of silicon oxygen tetrahedron chain. In the dry state, the Ca and Si atoms in the C–S–H molecule are superposed partially, the interlayer zones are reduced, the whole molecular structure system becomes compact, the atoms are arranged closely with evident disordering, and the fracture process is easy to cause movement of dislocations, as shown in Figure 10a, so dry concrete has good ductility. With the increase in the water saturation of concrete, most water molecules enter the interlayer zones of C–S–H, and a small number of water molecules are dispersed in the silicon oxygen tetrahedron chain and calcium oxygen laminar zones [33], the $O_S$ (oxygen atom in the C–S–H molecule) in $Ca_W$–$O_S$ and $Ca_S$–$O_S$ is partially replaced by the oxygen atoms $O_W$ in water molecules to form $Ca_W$–$O_W$ and $Ca_S$–$O_W$, the single-layer $O_S$–$H_W$ turns into double- or multi-layer Os–$H_W$ and $O_W$–$H_W$, the interlayer zone in the whole molecular system becomes evident, the atom superposition weakens, the gap among atoms enlarges, making it hard to cause movement of dislocations, and the deformation amount decreases when the chemical bonds break, as shown in Figure 10b,c, so the brittle characteristic of wet concrete is significant. Since $Ca_W$–$O_W$, $Ca_S$–$O_W$, and $O_W$–$H_W$ have larger bond lengths than $Ca_W$–$O_S$, $Ca_S$–$O_S$, and $O_S$–Hw, after water molecules enter the C–S–H molecular system, some chemical bonds have an increase in length and reduction in strength, the whole molecular system becomes loose, and its stability declines, so that the strength and fracture toughness of macroscopic concrete decrease with increasing water saturation.

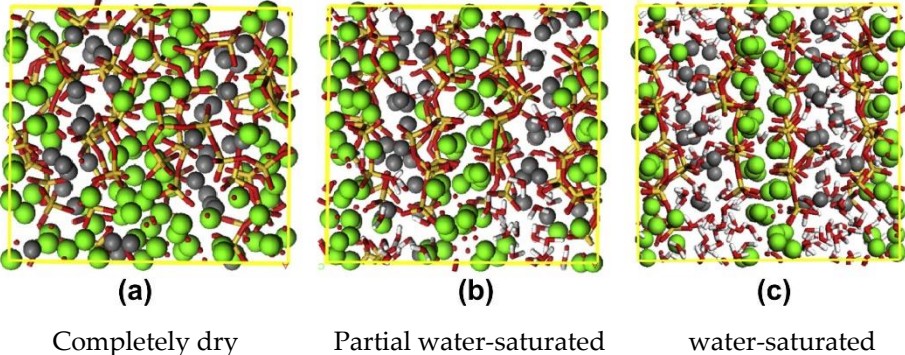

| **(a)** | **(b)** | **(c)** |
| Completely dry | Partial water-saturated | water-saturated |

**Figure 10.** Snapshots of the molecular structures of C–S–H gel samples from the dry state to the saturated state. Note: Yellow and red bonds represent the silicate chain (Si–Os); the gray and green balls correspond to the interlayer calcium atoms (Caw) and the layered calcium atoms (Cas), respectively. The red and white sticks represent water molecules.

## 5. Conclusions

(1) For all experimental groups, water saturation increases with the increase in immersion time, and the higher water to cement ratio, the faster the increase speed of the water saturation. The water absorption process of concrete is roughly divided into three stages, i.e., linear water absorption, nonlinear water absorption, and approximate saturation.

(2) The compressive strength, splitting tensile strength, and fracture toughness of concrete exhibit an approximately linearly decreasing trend with the increase in water saturation, and the mechanical properties of concrete with a high water cement ratio have higher sensitivity to water saturation than those of concrete with a low water cement ratio.

(3) The higher the water saturation of concrete, the larger the slope of the ascending part of the uniaxial compressive stress-strain curve, and the smaller the peak strain corresponding to the peak compressive stress, in addition, both $COMD_C$ and $Y_C$ decrease in this case.

**Author Contributions:** Data curation, X.L.; Formal analysis, Z.L.; Funding acquisition, G.Z.; Methodology, G.Z. and Z.L.; Resources, G.Z. and Z.L.; Software, X.L.; Validation, X.L.

**Funding:** This research was supported by the Scientific Research Foundation of Kunming University of Science and Technology (KKSY201704026), Supported by Open Research Fund Program of State key Laboratory of Hydroscience and Engineering (sklhse-2019-C-02).

**Acknowledgments:** The authors would like to thank the anonymous reviewers for their constructive suggestions to improve the quality of the paper.

**Conflicts of Interest:** The authors declare no conflict of interest.

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
