# Peer review of "Experimental Study on Static Mechanical Properties and Moisture Contents of Concrete Under Water Environment"

_sustainability, doi:10.3390/su11102962_

Round 1

Reviewer 1 Report

The topic is interesting as a moisture content in concrete does have a significant effect on its  mechanical properties. However, the presentation of results must be significantly improved, as is English language. Detailed remarks are as follows:

Water/cement ratio is the primary factor affecting concrete mechanical properties a the results should be discussed with reference to w/c;

Table 1 – please add the w/c

…in accordance with Chinese 92 standard GB175-2007, …according to Chinese 100 standard DL/T5330-2005. – please describe the regulations; they are not widely known

C15, C20, C30 – describe the correlation with the strength grade and correlation with Eurocode concreto class

2.2.1 Water saturation test procedure and 2.2.2 Mechanical properties test procedure - the description of the tests should be more clear, now we can get lost in the description of the number of samples

Fig.1 – add the the supports and loads and the real dimensions

Fig.2 and Fig.3 are unnecessary, they bring nothing new

Chapter 3.2 : … statistical analysis was made for the compressive strength… - there is no statistical analysis in the article

Fig.6 and Fig.7 – please present all data (3 for 1 sample), not the mean values

Eq.2 and 3, Si is not described

Please discuss possible general application of the proposed equations, e.g. for concretes of other classes

Fig.8  - the results are only for C20; it will be interesting to compare all curves for C15,C20 and C30 as well as with different w/c; only after adding these tests the article will be complete

Fig.10, 11 – the same comment as above

Please comment the results for tensile strength; some works suggests no significant effect of moisture content on splitting tensile strength

Author Response

For images cannot be inserted, we are very sorry. Please check the attached modification instructions (word in the attachment )

Response to Reviewer 1 Comments

Dear reviewer

Thank you for the comments concerning our manuscript entitled “Experimental research on drying control condition with minimal effect on concrete strength” (ID: sustainability-474117). Those comments are all valuable and very helpful for revising and improving our paper, as well as the important guiding significance to our researches. We have studied comments carefully and have made correction which we hope meet with approval. Revised portion are marked in red in the paper. The main corrections in the paper and the responds to your comments are as flowing. If any problem still exists in our manuscript, we will cooperate with you and reviewers to revise our manuscript until the manuscript meet the requirements.

Point 1: Water/cement ratio is the primary factor affecting concrete mechanical properties the results should be discussed with reference to w/c. Table 1 – please add the w/c.

Response 1: Your suggestions are very constructive for our research, and we have made correction according to your comments. The results reference to w/c are added in Discussions of this manuscript, and a new discussion for content is as flow, and the w/c of concrete are added in Table 1.

Line103:

Table 1. Mix proportion of concrete

Strength   grade

          Mix   proportion /(kg·m-3)

China

European

w/c

Water

Cement

Fine  ggregate

   Coarse aggregate

C15

C12

0.65

158

243

729

   1418

C20

C16

0.55

150

273

615

   1370

C30

C25

0.42

165

391

581

   1292

Point 2: in accordance with Chinese 92 standard GB175-2007, …according to Chinese 100 standard DL/T5330-2005. – please describe the regulations; they are not widely known

Response 2: Your suggestions are very constructive for our research, and we have made correction according to your comments. The Chinese standard GB175-2007 and DL/T5330-2005 all are be explained by references [21][22], and our details are as flowing.

Line100 and Line188:

[21] Chinese standard. GB175–2007. Common Portland Cement.

[22] Chinese standard. DL/T5330-2005. Code for mix design of hydraulic concrete.

Point 3: C15, C20, C30 – describe the correlation with the strength grade and correlation with Euro code concrete class

Response 3:  According to the Chinese standard (SL352-2006 Hydraulic Concrete Test Procedure), the specimens of the concrete compressive strength is cube specimen 150mm×150 mm×150mm. However , the specimens of the concrete compressive strength is Cylindrical specimen Φ150mm×300 mm with European concrete code . The corresponding relation of concrete strength grade between China and Europe was described in the following table.

Table 2. Comparison table of concrete strength grades in China with those in Europe

China

(Cube specimen)

C15

C20

C25

C30

C35

C40

C45

C50

C55

C60

Europe (Cylindrical specimen)

C12

C16

C20

C25

C30

C35

C40

C45

C50

Point 4: 2.2.1 Water saturation test procedure and 2.2.2 Mechanical properties test procedure - the description of the tests should be more clear, now we can get lost in the description of the number of samples.

Response 4: The Tests for compressive strength, split tensile strength were carried out in accordance with the Chinese standard SL352-2006 (Hydraulic Concrete Test Procedure). The Tests for fracture toughness was carried out in accordance with the Chinese standard DL/T5332-2005 (norm for fracture test of hydraulic concrete). The methodology of the test was added in the manuscript, and our details are as flowing.

Line185:

The loading rate was 0.3 MPa/s for test of compressive strength and 0.05 MPa/s for test of splitting tensile strength, and the concrete strength test was conducted according to the Hydraulic Concrete Test Procedure (SL352-2006) [23]

Line195:

In the three-point bending beam test, equal displacement rate was used to control loading and the loading rate was 0.1 mm/min; the test process was conducted strictly according to the Hydraulic Concrete Fracture Test Procedure (DL/T5332-2005) [24].

 Line143:

The number of samples of compressive strength, rupture tensile strength, fracture toughness has been described in Table 3.

Table 3. Specific content of each group

Specimen   type

Amount

Size/mm

Text   item

Standard   cube

3

150×150×150

Compressive strength

Standard   cube

3

150×150×150

Splitting tensile strength

Three-point bend beam

5

100×100×515

Fracture toughness

Summation

11

Point 6: Fig.1 – add the supports and loads and the real dimensions

Response 6: The supports, loads and the real dimensions were added to figure 1.

Figure 1. Configuration of three-point bending precracked beam

Point 7: Fig.2 and Fig.3 are unnecessary, they bring nothing new

Response 7: Your suggestions are very constructive for our research, and we have made correction according to your comments. Fig.2 and Fig.3 are unnecessary, and Fig.2 and Fig.3 were removed in the manuscript.

Point 8: Chapter 3.2 : … statistical analysis was made for the compressive strength… - there is no statistical analysis in the article

Response 8: The compressive strength of concrete was statistically analyzed in Fig. 6 and equation 2 in the manuscript.

The compressive strength of C15, C20, and C30 concrete specimens exhibit an approximately linearly decreasing trend with the increase in water saturation. Given the same water saturation, the lower the strength grade of concrete, the larger the decrease magnitudes of compressive strength. For example, when all concrete specimens were in water saturated state, the compressive strengths of C15, C20, and C30 concrete specimens decreased by 40.08%, 34.65%, and 32.23%, respectively, compared with the experimental groups in dry state.

Point 9: Fig.6 and Fig.7 – please present all data (3 for 1 sample), not the mean values

Response 9:  We are very sorry for our negligence of Fig.6 and Fig.7, presenting all data (3 for 1 sample) was more reasonable and scientific. 

Fig.  The change of tensile strength of concrete under different saturation

As can be seen from the above picture, the content of the Fig is rather confusing. It should be emphasized that these data from Fig.6 and Fig.7 are not simple arithmetic averages, and the data is processed as follows:

As stipulated in the Chinese national standard SL352-2006 (Hydraulic Concrete Test Procedure), the compressive strength should be calculated as the arithmetic mean of the measured value of three specimens. Besides, in case that the difference between maximum or minimum compressive strength of one specimen and intermediate compressive strength of three specimens is 15% over the intermediate compressive strength, the intermediate compressive strength of these three specimens should be taken as the value of the compressive strength; The test will be invalid in the event of two specimens obeying the above provision, at this time, this experiment must be repeated. The relative compressive strength is defined as the ratio of concrete compressive strength under different drying conditions to that of under the standard condition.

We have added the method of data processing to the manuscript.

Point 10: q.2 and 3, Si is not described

Response 10: Si of q.2 and 3 was described in q.1, and we have made correction according to your comments. Si of q.2 and 3 was described in the manuscript, and our details are as flowing:

Line269: Si   denotes the modulus water saturation of concrete (%).

Point 11: Please discuss possible general application of the proposed equations, e.g. for concretes of other classes

Response 11: This suggestion provides a good guide for our follow-up study. The extension of prediction equations for compressive strength and splitting tensile strength of concrete relative to water saturation were C15, C20, C30, For example, prediction equations (2) and (3). The mechanical properties of low strength grade concrete are more sensitive to humidity. According to our current research content, we can only qualitatively predict the change rule of mechanical properties of concrete with other strength grades. At present, we are carrying out physical experimental research on other strength grades of concrete. In our subsequent research results, we will systematically study the mechanical properties of concrete with C35, C40, C45, C50 and other strength grades.

Point 12: Fig.8- the results are only for C20; it will be interesting to compare all curves for C15, C20 and C30 as well as with different w/c; only after adding these tests the article will be complete.

Response 12: thank you for your kind suggestion. This paper presents an experiment to investigate the influence of immersion time on the moisture content of concrete, the influence of moisture on the Compressive strength, splitting tensile strength, fracture toughness. The research content is systematic and the number of physical test samples is large. Because of the limitations of the article, this manuscript aims at the expression of experimental laws and the explanation of the mechanism of experimental phenomena. Physical experiments on Uniaxial compression stress-strain relationship of C15, C20 are being studied, and we will analyze these results in depth. Research papers will be written on concrete stress-strain relationships affected the moisture content.

Point 13: Fig.10, 11 – the same comment as above

Response 13: We are very sorry for our negligence of the Fig.10, 11 – the same comment.

Figure 10 illustrates the fracture mouth opening displacement of concrete with different humidity along with the load change. Figure 11 illustrates the vertical displacement of loading point of concrete with different humidity along with the load change. The comment of Fig.10 and Fig.11 were completely different, and relevant revisions have been noted in the manuscript.

Point 14: Please comment the results for tensile strength; some works suggests no significant effect of moisture content on splitting tensile strength

Response 14:  The reviewer is very familiar with the research content, and few research content suggests no significant effect of moisture content on splitting tensile strength. Most of the research results show that humidity also has an effect on the tensile strength of concrete fracture, and concrete compression damage is essentially tensile damage. The decrease in tensile strength of wet concrete can be interpreted from the perspective of energy, the work done by external force during the generation, convergence and extension of cracks in concrete needs to overcome the surface energy forming microcracks. After water permeates into concrete, it reduces the Van der Waals forces between the microscopic particles of materials and weakens the cohesion among the particles of concrete so that the surface energy is reduced, and then the energy needed to form new fracture planes is decreased and the work done by external force for extension of initial cracks is less, macroscopically exhibiting decrease in tensile strength.

We tried our best to improve the manuscript and made some changes in the manuscript.  These changes will not influence the content and framework of the paper. We appreciate for your warm work earnestly, and hope that the correction will meet with approval.
    Once again, thank you very much for your comments and suggestions.

Yours sincerely,

Assistant professor Zhang

E-mail: bene@nwsuaf.edu.cn, Tel: +86 13669767383

Other authors:

Guohui Zhang, Tel: +86 15091199943, E-mail address: zgh_water@kmust.edu.cn.

Xiaohang Li, Tel: +86 19922102384, E-mail address: lxh@kmust.edu.cn.

Reviewer 2 Report

In general, the manuscript can be divided into two part: 

1) qualitative findings about the mechanical properties of the concrete materials under the moisture condition, and 

2) presenting a quantitative model for the results.

Strictly speaking, nearly all the reported results in part 1 are institutive. The response of the concrete in water environment is well known, and this part does not add any new information.

Results of part 2 is more interesting. However, all the experimental tests are very simple, they are limited, and generalizing the results and providing a big equation does not seem to be adequate. 

The authors need to expand their study with more tests, clear definition of the method, tabulation of the raw data as an appendix, and more state-of-the-art method for curve fitting and post-processing.

Author Response

Response to Reviewer 2 Comments

Dear reviewer

Thank you for your comments concerning our manuscript entitled “Experimental research on drying control condition with minimal effect on concrete strength” (ID: sustainability-474117). Those comments are all valuable and very helpful for revising and improving our paper, as well as the important guiding significance to our researches. We have studied comments carefully and have made correction which we hope meet with approval. Revised portion are marked in red in the paper. The main corrections in the paper and the responds to the your comments are as flowing. If any problem still exist in our manuscript, we will cooperate with you and reviewers to revise our manuscript until the manuscript meet the requirements.

Point

Comments and Suggestions for Authors

In general, the manuscript can be divided into two part:

1) qualitative findings about the mechanical properties of the concrete materials under the moisture condition, and

2) presenting a quantitative model for the results.

Strictly speaking, nearly all the reported results in part 1 are institutive. The response of the concrete in water environment is well known, and this part does not add any new information.

Results of part 2 is more interesting. However, all the experimental tests are very simple, they are limited, and generalizing the results and providing a big equation does not seem to be adequate.

The authors need to expand their study with more tests, clear definition of the method, tabulation of the raw data as an appendix, and more state-of-the-art method for curve fitting and post-processing.

Response:

thank you for your kind suggestion, it is my honor to share my work with peer researchers.

This paper presents an experiment to investigate the influence of immersion time on the moisture content of concrete, the influence of moisture on the Compressive strength, splitting tensile strength, fracture toughness. The research content is systematic and the number of physical test samples is large. Because of the limitations of the article, this manuscript aims at the expression of experimental laws and the explanation of the mechanism of experimental phenomena. however, the dataset we adopted in this current research was now deeply analyzed in another paper of my collegues and I. Besides, this paper is almost finished and will go through revision soon.

    According to your Suggestions for modification, we have made corresponding modifications in the manuscript, and the modified part has been standardized in red. The details of the changes are shown in the manuscript.

We tried our best to improve the manuscript and made some changes in the manuscript.  These changes will not influence the content and framework of the paper. We appreciate for your warm work earnestly, and hope that the correction will meet with approval.
        Once again, thank you very much for your comments and suggestions.

Yours sincerely,

Assistant professor Zhang

E-mail: bene@nwsuaf.edu.cn, Tel: +86 13669767383

Other authors:

Guohui Zhang, Tel: +86 15091199943, E-mail address: zgh_water@kmust.edu.cn.

Xiaohang Li, Tel: +86 19922102384, E-mail address: lxh@kmust.edu.cn.

Reviewer 3 Report

However, there are some minor grammar and spelling check are required. In the abstract, the author mentioned "compressive strength decreases by 40.08%, 36.08%, and 33.73%, respectively, splitting tensile 19 strength decreases by 45.39%, 42.61%, and 35.18%, respectively, and fracture toughness decreases 20 by 57.31%, 49.92%, and 46.76%, respectively". However, in the content, the author (s) didn't discuss the methodology of the test. The author must mention the standard that these tests are in compliance with, for instance ASTM XXXX or ACI XXX,... otherwise the procedure should be stated clearly. For fracture toughness, there are variety of methods, but it is not clear what method was used in this experiment. The results could reveal more information for reader. I believe the results part is too wordy and could be organized in a better way. 

Author Response

Response to Reviewer 3 Comments

Dear reviewer

Thank you for the your comments concerning our manuscript entitled “Experimental research on drying control condition with minimal effect on concrete strength” (ID: sustainability-474117). Those comments are all valuable and very helpful for revising and improving our paper, as well as the important guiding significance to our researches. We have studied comments carefully and have made correction which we hope meet with approval. Revised portion are marked in red in the paper. The main corrections in the paper and the responds to the your comments are as flowing. If any problem still exist in our manuscript, we will cooperate with you and reviewers to revise our manuscript until the manuscript meet the requirements.

Point 1: However, there are some minor grammar and spelling check are required. otherwise the procedure should be stated clearly.

Response 1: Your suggestions are very constructive for our research, and we have made correction according to your comments. The minor grammar and spelling check were modified in the manuscript.

Point 2: In the abstract, the author mentioned "compressive strength decreases by 40.08%, 36.08%, and 33.73%, respectively, splitting tensile 19 strength decreases by 45.39%, 42.61%, and 35.18%, respectively, and fracture toughness decreases 20 by 57.31%, 49.92%, and 46.76%, respectively". However, in the content, the author (s) didn't discuss the methodology of the test. The author must mention the standard that these tests are in compliance with, for instance ASTM XXXX or ACI XXX,...

Response 2: The Tests for compressive strength, split tensile strength were carried out in accordance with the Chinese standard SL352-2006 (Hydraulic Concrete Test Procedure). The Tests for fracture toughness was carried out in accordance with the Chinese standard DL/T5332-2005 (norm for fracture test of hydraulic concrete). The methodology of the test was added in the manuscript, and our details are as flowing.

Line187:

The loading rate was 0.3 MPa/s for test of compressive strength and 0.05 MPa/s for test of splitting tensile strength, and the concrete strength test was conducted according to the Hydraulic Concrete Test Procedure (SL352-2006) [23]

Line198:

In the three-point bending beam test, equal displacement rate was used to control loading and the loading rate was 0.1 mm/min; the test process was conducted strictly according to the Hydraulic Concrete Fracture Test Procedure (DL/T5332-2005) [24].

Point 3: For fracture toughness, there are variety of methods, but it is not clear what method was used in this experiment.

Response 3: For fracture toughness, there are variety of methods, fracture toughness test was carried out by fracture test of three-point bending beam according to the Hydraulic Concrete Fracture Test Procedure (DL/T5332-2005), and the three-point bending beam test of concrete was carried out using a microcomputer controlled electro-hydraulic servo universal tester. The testing for fracture toughness of methods has been described in the manuscript, and our details are as flowing.

Line188:

The fracture toughness test was carried out by fracture test of three-point bending beam, and the three-point bending beam test of concrete was carried out using a microcomputer controlled electro-hydraulic servo universal tester, whose model number was SHT4305 and whose maximum test pressure was 300 kN. The load and displacement were measured using load cells and displacement transducers, respectively, the crack mouth opening displacement (CMOD) was measured using a clip-on extensometer, and a computer acquisition system connected with the tester was used to record and display data, and to automatically record the whole process curves of load versus loading point deflection and of load versus CMOD in real time

Point 4: The results could reveal more information for reader. I believe the results part is too wordy and could be organized in a better way.

Response 4: Your suggestions are very constructive for our research, the results part was truly too wordy, and we have made correction according to your comments. The conclusion section has been rewritten and simplified and a new discussion for content is as flow.

Conclusions

(1) For all experimental groups, water saturation increases with the increase in immersion time, and the higher water cement ratio, the faster the increase speed of the water saturation. The water absorption process of concrete is roughly divided into three stages, i.e., linear water absorption, nonlinear water absorption, and approximate saturation.

(2) The compressive strength, splitting tensile strength, and fracture toughness of concrete exhibit an approximately linearly decreasing trend with the increase in water saturation, and the mechanical properties of concrete with a high water cement ratio have higher sensitivity to water saturation than those of concrete with a low water cement ratio.

        (3) The higher the water saturation of concrete, the larger the slope of the ascending part of the uniaxial compressive stress-strain curve, and the smaller the peak strain corresponding to the peak compressive stress, in addition, both COMDC and YC decrease in this case.

We tried our best to improve the manuscript and made some changes in the manuscript.  These changes will not influence the content and framework of the paper. We appreciate for your warm work earnestly, and hope that the correction will meet with approval.
    Once again, thank you very much for your comments and suggestions.

Yours sincerely,

Assistant professor Zhang

E-mail: bene@nwsuaf.edu.cn, Tel: +86 13669767383

Other authors:

Guohui Zhang, Tel: +86 15091199943, E-mail address: zgh_water@kmust.edu.cn.

Xiaohang Li, Tel: +86 19922102384, E-mail address: lxh@kmust.edu.cn.

Round 2

Reviewer 1 Report

I accept the revised version of the article

Author Response

Thank you so much for your support 

Reviewer 2 Report

My concern is not resolved. The authors did not any particular change based on my comment.

Author Response

Dear reviewer

Thank you for your comments concerning our manuscript entitled “Experimental research on drying control condition with minimal effect on concrete strength” (ID: sustainability-474117). Those comments are all valuable and very helpful for revising and improving our paper, as well as the important guiding significance to our researches. We have studied comments carefully and have made correction which we hope meet with approval. Revised portion are marked in red in the paper. The main corrections in the paper and the responds to the your comments are as flowing. If any problem still exist in our manuscript, we will cooperate with you and reviewers to revise our manuscript until the manuscript meet the requirements.

Point 1In general, the manuscript can be divided into two part:

1) qualitative findings about the mechanical properties of the concrete materials under the moisture condition, and

2) presenting a quantitative model for the results.

Response 1: Thank you for your kind suggestion, it is my honor to share my work with peer researchers. My manuscript really should be divided into two parts, the first part is the qualitative description of the experimental results, the second part should present a quantitative model for the results. This paper presents an experiment to investigate the influence of immersion time on the moisture content of concrete, the influence of moisture on the Compressive strength, splitting tensile strength, fracture toughness. The research content is systematic and the number of physical test samples is large. Because of the limitations of the article, it is very hard to construct predictive mathematical models for moisture content and compressive strength, splitting tensile strength, fracture toughness. In this paper, the mechanism of the experimental results was explained mainly from the microscopic scale in discussion.

Point 2Strictly speaking, nearly all the reported results in part 1 are institutive. The response of the concrete in water environment is well known, and this part does not add any new information.

Response 1: The response of the concrete in water environment is well known, the systematic study of concrete moisture content is a necessary prerequisite for the study of mechanical properties of wet concrete. In order to achieve different moisture content of concrete, different researchers use different drying methods when drying concrete specimens to determine the moisture content, and the difference in strength due to different drying methods will inevitably interfere with the evaluation of the degree of influence of moisture content on the strength of concrete. This research team carried out study on the influence of drying condition on the strength of concrete, and got the drying process with the minimum damage to concrete. Based on this drying method, the free water absorption process of concrete was studied first in this paper to generate concrete specimens with different moisture contents.  

The manuscript is based on the moisture content of concrete obtained from the following references

Zhang Guohui, Li Zongli, Zhang Linfei, Shang Yujuan, Wang Hang. Experimental research on drying control condition with minimal effect on concrete strength, Construction and Building Materials,135(2017),194-202.

Point 3Results of part 2 is more interesting. However, all the experimental tests are very simple, they are limited, and generalizing the results and providing a big equation does not seem to be adequate. The authors need to expand their study with more tests, clear definition of the method, tabulation of the raw data as an appendix, and more state-of-the-art method for curve fitting and post-processing.

Response 1: The research team is building a strength prediction model based on the mesoscopic inclusion theory, the dataset we adopted in this current research was now deeply analyzed in another paper of my collegues and I. Besides, this paper is almost finished and will go through revision soon. Your suggestion of tabulation of the raw data as an appendix is perfectly reasonable.

The original data: nondestructive drying concrete saturation and   soaking time 
Immersion   time/hC15 concrete   saturation/MPaC20 concrete   saturation/MPaC20 concrete   saturation/MPa
0.5024.56117.07019.195
1.0030.79921.69023.969
1.5036.45425.70528.743
2.0041.62129.62131.891
2.5046.20133.33635.751
3.0049.80836.24838.493
3.5053.51339.16040.931
4.0056.24141.36842.962
4.5059.75044.18045.602
5.0062.38146.18847.431
5.5065.20848.39749.056
6.0066.76849.70150.274
6.5069.20351.91052.407
7.0071.15453.51753.321
7.5072.42054.52154.539
8.0075.05356.62956.266
8.5076.32057.83457.383
9.0078.07259.13958.906
9.5079.34060.44459.516
10.0080.80161.54960.430
11.0083.33663.95862.360
12.0085.87066.66964.289
13.0087.62368.47666.219
14.0089.18370.18267.438
15.0090.54671.98969.063
16.0091.91173.99770.282
17.0092.88575.60471.907
19.0093.95778.51575.154
21.0094.54181.12576.477
26.0094.93387.04781.150
31.0095.61490.46285.719
35.0096.10192.36887.548
47.0096.29894.57890.699
54.0096.88395.68393.945
59.0096.97996.18495.773
71.0096.88296.58697.702
77.0096.97996.68798.412
95.0097.46797.28998.593
103.0097.46797.39098.787
120.0097.75897.59099.054
131.0097.75997.89299.224
143.0098.05197.89299.321
165.0098.34097.69299.418
185.0098.92998.99699.466
222.0099.02599.19799.563
268.00100.000100.000100.000

The original data: Compressive strength at different saturation
saturation/%Compressive strength   of C15 concrete/MPasaturation/%Compressive strength   of C20 concrete/MPasaturation/%Compressive strength   of C30 concrete/MPa
0.00025.1570.00027.3000.00035.940
49.80820.67736.24824.45238.49331.500
80.80117.78161.54921.73460.43027.530
94.93316.86687.04720.37081.49725.451
96.88215.22396.58619.60497.70224.632
100.00015.073100.00017.451100.00023.817

The original data: Tensile strength of splitting at different   saturation
saturation/%Tensile strength of   splitting of C15 concrete/MPasaturation/%Tensile strength of   splitting of C20 concrete/MPasaturation/%Tensile strength of   splitting of C30 concrete/MPa
0.0001.7610.0001.8460.0002.033
49.8081.28136.2481.72238.4931.633
80.8011.21361.5491.49760.4301.463
94.9331.02487.0471.34281.4971.365
96.8820.98396.5861.26197.7021.332
100.0000.962100.0001.059100.0001.318

The original data:Fracture toughness and maximum load at   different saturation
C15C20C30
saturationPmaxKICsaturationPmaxKICsaturationPmaxKIC
NMPa·m1/2NMPa·m1/2NMPa·m1/2
0.00015000.5060.00019090.6430.00022030.742
57.76510470.35350.00114350.48441.24518670.629
61.1629390.31662.97113470.45460.12316670.562
64.0697600.25688.00411470.38765.25414160.477
80.5657170.24295.89410580.35790.41312690.428
100.0006420.216100.0009560.322100.00011720.395

Data processing instructions:

As stipulated in the Chinese national standard SL352-2006 (Hydraulic Concrete Test Procedure), the compressive strength, tensile strength of splitting, fracture toughness should be calculated as the arithmetic mean of the measured value of three specimens. Besides, in case that the difference between maximum or minimum compressive strength of one specimen and intermediate compressive strength of three specimens is 15% over the intermediate compressive strength, the intermediate compressive strength of these three specimens should be taken as the value of the compressive strength; The test will be invalid in the event of two specimens obeying the above provision, at this time, this experiment must be repeated. The relative compressive strength is defined as the ratio of concrete compressive strength under different drying conditions to that of under the standard condition.

We tried our best to improve the manuscript and made some changes in the manuscript.  These changes will not influence the content and framework of the paper. We appreciate for your warm work earnestly, and hope that the correction will meet with approval.

        Once again, thank you very much for your comments and suggestions.

Yours sincerely,

Assistant professor Zhang

E-mail: bene@nwsuaf.edu.cn, Tel: +86 13669767383

Other authors:

Guohui Zhang, Tel: +86 15091199943, E-mail address: zgh_water@kmust.edu.cn.

Xiaohang Li, Tel: +86 19922102384, E-mail address: lxh@kmust.edu.cn.